# Hypervolemia in Dialysis Patients Impairs STAT3 Signaling and Upregulates miR-142-3p: Effects on IL-10 and IL-6

**DOI:** 10.3390/ijms25073719

**Published:** 2024-03-27

**Authors:** Christof Ulrich, Roman Fiedler, Eva Herberger, Zeynep Canim, Silke Markau, Matthias Girndt

**Affiliations:** 1Department of Internal Medicine II, Martin Luther University Halle-Wittenberg, 06120 Halle (Saale), Germany; roman.fiedler@uk-halle.de (R.F.); silke.markau@uk-halle.de (S.M.); matthias.girndt@uk-halle.de (M.G.); 2KfH Nierenzentrum, 06120 Halle (Saale), Germany

**Keywords:** chronic kidney failure, hypervolemia, hemodialysis, IL-10, STAT3, IL-6, SOCS3, miR-142, inflammation

## Abstract

Fluid overload in hemodialysis patients (HD) has been proven to be associated with inflammation. Elevated levels of the pro-inflammatory cytokine interleukin-6 (IL-6) appear to be inadequately counterbalanced by the anti-inflammatory cytokine interleukin-10 (IL-10). We initiated a cross-sectional study enrolling 40 HD patients who were categorized by a bioimpedance measurement in normovolemic (N; 23) and hypervolemic (H; 17) groups to test whether IL-10- and IL-6-related signal transduction pathways (signal transducer of transcript 3: STAT3) and/or a post-transcriptional regulating mechanism (miR-142) are impaired by hypervolemia. IL-10/IL-6 transcript and protein production by PBMCs (peripheral blood mononuclear cells) were determined. Phospho-flow cytometry was used to detect the phosphorylated forms of STAT3 (pY705 and pS727). miR-142-3p/5p levels were detected by qPCR. Hypervolemic patients were older, more frequently had diabetes, and showed higher CRP levels. IL-10 transcripts were elevated in H patients but not IL-10 protein levels. In spite of the elevated mRNA expression of the suppressor of cytokine expression 3 (SOCS3), IL-6 mRNA and protein expression were increased in immune cells of H patients. The percentage of cells staining positive for STAT3 (pY705) were comparable in both groups; in STAT3 (pS727), however, the signal needed for full transactivation was decreased in H patients. miR-142-3p, a proven target of IL-10 and IL-6, was significantly elevated in H patients. Insufficient phosphorylation of STAT3 may impair inflammatory and anti-inflammatory cytokine signaling. How far degradative mechanisms induced by elevated miR-142-3p levels contribute to an inefficient anti-inflammatory IL-10 signaling remains elusive.

## 1. Introduction

Fluid imbalance is a serious problem in chronic kidney disease patients treated with maintenance dialysis (HD). It has been known that, notably, fluid excess (hypervolemia) is pertinent to the clinical outcome. On the one hand, overhydration is linked to hypertension and left ventricular hypertrophy; on the other hand, it was proven that the removal of fluid overload is associated with the improvement of blood pressure and cardiac remodeling [1]. The mechanism by which hypervolemia contributes to cardiovascular diseases involves inflammatory cytokines and the complex interplay between pro-inflammatory and anti-inflammatory cytokines. The correct induction of an inflammatory response needs to be followed by a timely shut-down of the inflammatory insult. In the setting of chronic inflammation, however, the inflammatory power appears to predominate the anti-inflammatory response. Therefore, the inflammatory burden of hypervolemic chronic kidney disease patients is very high. Several authors reported higher IL-6 serum levels in overhydrated dialysis patients [2,3,4]. Further, it is widely acknowledged that IL-6 is linked to cardiovascular diseases, in particular to chronic coronary syndrome [5], atherosclerosis [6], and hypertension [7] in chronic kidney disease.

In addition, high numbers of circulating monocytes and tissue macrophages aggravate both hypertension and target organ damage. Administration of the anti-inflammatory cytokine IL-10 can restrain the production of pro-inflammatory cytokines and chemokines during hypertension [8].

This study is intended to describe translational and post-translational mechanisms of cytokine induction by overhydration. The induction of inflammatory events requires certain intracellular signaling pathways. Signaling via the transduction and the activator of the transcription (STAT) 3 pathway is essential for the induction of the anti-inflammatory IL-10; however, STAT3 can also be induced by the pro-inflammatory cytokine IL-6 [9]. STAT3 activation is associated with different phosphorylation steps in tyrosine and serine residues of the molecule [10,11,12]. The cytokine network is further influenced by another class of proteins, namely the suppressors of cytokine signaling (SOCS). As suggested by Ding and coworkers, SOCS1 and SOCS3 may partially inhibit IL-10-mediated STAT3 effects [13].

It is of note that the aryl hydrocarbon receptor (AhR) seems to be involved in central anti-inflammatory mechanisms. AhR is a ligand-activated transcription factor that was initially discovered as a system mediating dioxin-related effects, but newer studies have shown that AhR is a transcription factor that integrates not only environmental but also microbial and metabolic signals [14]. AhR is known to be activated in chronic kidney disease [15]. Interestingly, the loss of AhR is associated with reduced IL-10 expression in lipopolysaccharide (LPS)-stimulated macrophages [16].

To study putative mechanisms that translationally or post-transcriptionally regulate IL-10, we investigated the translational repressor protein 4E-BP1. The eukaryotic translation factor eIF4E is blocked by a reversible association between eIF4E and the repressor-binding proteins 4E-BP1, 4E-BP2, and 4E-BP3. The phosphorylation of 4E-BP1 (p4E-BP1) and 4E-BP2 (p4E-BP2) induces the dissociation from eIF4E, thus allowing the translation of proteins [17]. William and colleagues demonstrated that 4E-BP1 and 4E-BP2 are also able to influence IL-10 production via this mechanism. Studying the phosphorylated form of 4E-BP1 can give a clue to general translational repression mechanisms induced by overhydration [18]. Not only translational repressors but also micro-RNAs (miRs) have proven to be powerful regulators of gene expression. miRs, small, 21–25 nucleotide non-coding RNAs, regulate the expression of their targets by binding to the 3′-untranslated regions (3′-UTR) of the corresponding mRNA in order to translationally repress or degrade the product. miR-142 is specifically expressed by hematopoietic cells and has an impact on IL-6 expression [19]. Bioinformatic analysis revealed that IL-10 is also a predicted target of miR-142 [20]. Interestingly, miR-142-3p levels are inversely correlated with IL-10 in patients with systemic lupus erythematosus. Thus, the down-regulation or knock-out of miR-142-3p is associated with increased levels of IL-10 in CD4-positive lymphocytes [20] and vice versa.

We therefore investigated the roles of IL-10 and IL-6 in normo- and hypervolemic patients, trying to figure out regulatory mechanisms to gain insight about the altered anti-inflammatory response of overhydrated hemodialysis patients.

## 2. Results

The patient characteristics of this cohort were reported recently [21]. In short, the hydration status of the patients was analyzed by bioelectric impedance vector analysis (BIVA). Hypervolemic patients (n = 17) had an average phase angle of 3.8 ± 0.8°, whereas normovolemic patients (n = 23) showed a phase angle of 5.2 ± 1.2° (*p* < 0.001). Further, H patients were older (H: 67.5 ± 10.2, *p* = 0.020 vs. N: 57.6 ± 14.2) and more frequently had diabetes (H: 52.9% vs. N:17.4%, *p* = 0.038). Inflammation, as measured by hsCRP assay, was significantly elevated in H patients (H: 14.2 ± 16.1 vs. N: 5.2 ± 6.3, *p* = 0.020).

### 2.1. mRNA Expression of IL-10 Is Elevated in PBMCs of H Patients

PBMCs of H patients constitutively expressed higher levels of IL-10 mRNA compared to N patients (Figure 1a). It is known that LPS induces both inflammatory mediators and anti-inflammatory IL-10. Thus, PBMCs were challenged with 25 ng/mL LPS. All patients increased their IL-10 mRNA transcription (Figure 1b,c). The presence of an antagonist of the AhR pathway (AhR-A) during the stimulation of PBMCs with LPS decreased IL-10 transcripts to some extent in both patient groups (Figure 1b,c). The transcripts of the main target genes of the AhR pathway (Cyp1A and Cyp1B) were not differently expressed in N and H patients (Figure 1d,f). Cyp1A and Cyp1B mRNA expressions were downregulated upon LPS stimulation in both groups (Figure 1e,g). This effect was enhanced upon stimulation of the cells with LPS and AhR-A (Figure 1e,g). In contrast to IL-10, the inflammatory gene IL-6 was significantly upregulated when the AhR pathway was inhibited by AhR-A treatment (Figure 1h).

### 2.2. Frequency and Density Expression of IL-10 Are Not Different between N and H Patients

IL-10 protein secretion capacity in monocytes (CD14+) and lymphocytes (CD3+) was analyzed by intracellular FACS staining. Neither the frequency of IL-10-producing monocytes (Figure 2a) nor the mean IL-10 expression per cell (MFI, Figure 2d) was different between both groups. The frequency of monocytic cells staining positive for IL-10 was only slightly enhanced by LPS stimulation (Figure 2b,c). Additional challenge with an AhR-A had no effect on cells in comparison to PBMCs that were only stimulated with LPS (Figure 2b,c). Unlike the frequency of IL-10-producing monocytes, the median expression of IL-10 was marginally but significantly elevated in both patient groups upon LPS stimulation (Figure 2e,f). Regarding the lymphocytic IL-10 expression, we found no difference in both groups (Figure 2g). LPS as a prototypical stimulus of monocytes even appeared to have an inhibitory effect on the frequency of IL-10-producing T cells (Figure 2h,i). Additionally, IL-10 serum protein levels were also not different in N and H patients (N: 1.2 [0.6–4.9] vs. H: 1.1 pg/mL [0.6–76.7], *p* = 0.412).

Analytical separation of the IL-10 response in different immune cells clearly shows that in our experimental setting, monocytes are the main producer of IL-10. Different stimulatory conditions are necessary to induce a higher lymphocytic IL-10 production.

To confirm that hypervolemia is linked to an inflammatory state in immune cells, we decided to determine IL-6. As the analysis of IL-6 could not be integrated in the IL-10 flow cytometric assay, we measured the pro-inflammatory cytokine in cell supernatants (PBMC) of unstimulated samples. As expected, IL-6 (pg/mL) was significantly increased in PBMCs of H patients (N: 23.6 ± 17.8 vs. H: 58.1 ± 46.6, *p* = 0.003).

### 2.3. The Frequency of p4E-PB1-Producing Immune Cells Is Not Different in N and H Patients

Rapid activation or suppression of protein synthesis is a common regulatory mechanism which allows immune cells to respond to inflammation. The translational repressor protein 4E-PB1 is part of the translational initiation factor 4F, which is activated by the dissociation of its phosphorylated binding proteins. Analyzing monocytic (CD86+) and lymphocytic (CD4+) cells, we could not find significant differences between normo- and hypervolemic HD patients (Table 1). LPS challenge increased the percentage of p4E-BP1-positive monocytes in both groups. Additional challenge with AhR-A had no effect (Table 1).

A similar analysis in T helper cells (CD4+) also revealed no significant difference between N and H patients regarding p4E-BP1 (Table 2). The percentage of p4E-BP1 could be significantly increased upon LPS stimulation. Again, the combination of LPS/AhR-A did not lead to different results when compared to cells only stimulated with LPS (Table 2). The high inter-individual variability may indicate that factors other than fluid overload may be responsible for the variation in 4E-BP1 expression.

### 2.4. Lower Monocytic STAT3 (pS727) Activation in Hypervolemia

STAT3 has been proposed to bind and transactivate IL-10. For this process, STAT3 phosphorylation at tyrosine residue 705 (pY705) and at serine residue 727 (pS727) is essential. As can be seen in Figure 3a–c, STAT3 (pY705) is not different in both groups, neither at the basal level (Figure 3a) nor upon stimulation (Figure 3b,c). In contrast, the frequency of cells staining positive for STAT3 (pS727) is higher in normovolemic patients (Figure 3d). As opposed to STAT3 (pY705), the phosphorylation site at serine 727 is LPS-dependent (Figure 3e,f) in both patient groups. No additional effects were observed by dual stimulation with LPS and AhR-A (Figure 3e,f).

### 2.5. Hypervolemia Induces SOCS3 but Not SOCS1 mRNA Expression

As the suppressors of cytokine signaling (SOCS) are involved in the regulation of pro-inflammatory and anti-inflammatory cytokines, we analyzed the mRNA expressions of SOCS1 and SOCS3 in PBMCs of N and H patients. As demonstrated in Table 3, the transcription of the SOCS1 gene was not different between N and H patients, whereas the SOCS3 gene expression was significantly up-regulated in H patients.

### 2.6. miR-142-3p Expression Is Elevated in H Patients

miR-142 belongs to a group of miRs which can regulate cytokine expression in immune cells. It is of specific interest in our study because miR-142 appears to be involved in the regulation of both IL-10 and IL-6. As demonstrated in Figure 4a, miR-142-3p expression was significantly enhanced in PBMCs of H patients. Upon LPS stimulation, PBMCs of both groups up-regulated miR-142, and this effect was inhibited by the AhR-A challenge exclusively in H but not in N patients (Figure 4b,c).

Conversely to miR 42-3p, the expression of the second isoform of miR-142, mir-142-5p, was not statistically different in both groups, although there seemed to be a trend for higher miR-142-5p values in the H group, too. Further on, although not statistically significant, LPS stimulation resulted in slightly higher miR-142-5p values in both patient groups (Figure 4e,f).

## 3. Discussion

In spite of different limitations (patients with pacemakers, amputees, and patients with metal-on-metal prostheses, as well as decompensated cirrhosis patients, were excluded from analysis), bioimpedance analysis is assumed to be the gold standard for measuring volume overload in hemodialysis patients [22]. Several devices for monitoring fluid overload have been investigated in HD patients. These include the Fresenius Body Composition Monitor (Fresenius Medical Care, Bad Homburg, Germany) [22], Nikkiso DBB device (Nikkiso Medical Europe, Hamburg, Germany) [23], and the Nutriguard-M device (Data Input GmbH, Pöcking, Germany) [24]. The importance of bioimpedance analysis in dialysis patients is highlighted by the fact that bioimpedance parameters most probably represent independent predictors for mortality and cardiovascular events [25].

Our study supports data showing a close association between inflammation and fluid overload. Elevated levels of CRP mark the predominant inflammatory response in hypervolemic HD patients, compared to those with normovolemia [26]. Most likely, this is at least partially caused by imbalanced anti-inflammatory mechanisms. The most prominent member of the anti-inflammatory arsenal is IL-10, and in line with the theory—inflammation is followed by anti-inflammation—we confirmed an earlier observation indicating higher IL-10 mRNA expression in hypervolemia [4].

Recently, it was shown that the aryl hydrocarbon receptor pathway (AhR) has an immune modulatory function and is directly involved in the immune regulation of inflammatory macrophages [16,27]. AhR is a transcription factor that is typically activated by various organic environmental substances but can also be induced by LPS or tryptophan-derived metabolites [28]. The major targets of the AhR pathway are genes of the cytochrome P450 family. We speculated that tryptophane-derived uremic toxins engage the AhR and alter its downstream events. However, mRNA levels for cytochromes Cyp1A and Cyp1B as examples of these downstream events were not significantly different between our study groups. Therefore, there is no indication of a different AhR activation state in hypervolemic individuals. Further on, consistent with other reports, both genes are down-regulated in N and H patients upon LPS stimulation [29,30]. The anti-inflammatory nature of the AhR pathway is supported by our experiments using an AhR-antagonist: adding the antagonist to LPS-stimulated cells up-regulated IL-6 transcripts, while IL-10 transcripts tended to decrease. However, as this effect is only very small, we do not believe that AhR plays a major role in IL-10 signaling in hypervolemia.

While the mRNA expression of IL-10 was higher in the H group, the frequency of IL-10-producing monocytes and lymphocytes and the expression density of IL-10 were not different in N and H patients. This discrepancy of IL-10 mRNA and protein levels in hypervolemic patients was already observed earlier [4] and suggests a posttranscriptional mechanism being involved. Analyzing the repressor-binding protein 4E-BP1—a marker protein whose phosphorylated form is linked to de-repression of the translational factor eIF4E—we did not find differences in immune cells staining positive for this phosphor-protein between N and H patients. However, the analysis revealed remarkable interindividual differences. A polysome-based IL-10 analysis could be a specific means to study the impaired IL-10 response, an investigation which should be addressed in the future.

Next, we focused on the signal transduction pathway distal to the IL-10 receptor. The JAK (Janus kinase)/STAT3, a pathway which is involved in cell differentiation, apoptosis, and inflammation, is essential for all known functions of IL-10 [31], but paradoxically, not only IL-10 but also the pro-inflammatory cytokine IL-6 signals via STAT3 [9,32]. STAT3 has an important role in suppressing signal transduction mediated by toll-like receptors in phagocytic cells [33], thus preventing signal transduction via the transcription factor NF-kB [34]. Activation of STAT3 is linked to the phosphorylation at the tyrosine residue at position 705, but the full transactivation of STAT3 should only be achieved when, additionally, serine is post-translationally modified by phosphorylation at position 727. Thus, it is quite possible that an insufficient phosphorylation state of STAT3 impairs the quality of the anti-inflammatory response, i.e., the immune response in hypervolemia. Besides the activation of STAT3 signaling, IL-10 and IL-6 can also induce the expression of the regulatory element SOCS3. While SOCS3 clearly is involved in the repression of IL-6 by binding to the IL-6 receptor subunit gp130, the effects of SOCS3 on IL-10 signaling are controversial. Some authors state that IL-10 signal transduction is sensitive to SOCS3 inhibition [13,35]. Other study groups state that SOCS3 lacks affinity to the IL-10 receptor and thus has no effect on IL-10 signaling [36,37]. On the contrary, there are some indications that SOCS1, which is also upregulated by IL-10, may play a role as a feedback inhibitor for the transiency of the IL-10 signal [35]. In our study, we found elevated levels of SOCS3 transcripts in H vs. N patients, while SOCS1 mRNA expression was not different in both groups. Therefore, it seems possible that immune cells of H patients try to curtail the inflammatory IL-6 pathway by increasing the SOC3 mRNA levels. Nevertheless, IL-6 levels are still higher in H, compared to N patients. Therefore, anti-inflammatory compensation of immune cells in hypervolemic HD patients is insufficient to overcome the inflammatory load. Conceivably, the lack of an efficient STAT3 phosphorylation state hampers IL-10 and IL-6 signaling.

Besides the STAT3 pathway, micro-RNAs have proven to be powerful post-transcriptional regulators of IL-10 and IL-6. miR-142-3p/5p can directly target the 3′-untranslated region of IL-10, leading to gene repression [38]. In several studies, miR-143-3p expression was inversely associated with IL-6 or IL-10 levels. Therefore, Ding and colleagues impressively demonstrated that reduced miR-142-3p expression increases IL-10 protein levels in CD4-positive lymphocytes [20]. Gao and colleagues demonstrated that the overexpression of miR-142-3p in regulatory T cells resulted in a significant decrease in IL-10 and TGF-ß [39], while another group showed that the knock-down of miR-142 in dendritic cells is associated with enhanced expressions of IL-10 and IL-6 [40]. Conversely, the study of Fordham et al. found a direct relationship between miR-142-3p and inflammatory mediators. They showed that miR-143-3p was downregulated in macrophages and dendritic cells during differentiation, and this process was associated with the reduced release of IL-6 and TNF-α upon LPS stimulation [41]. Finally, Qing and colleagues measured elevated levels of miR-142-3p and TNF-α in chronic rhinosinusitis [42]. We found higher miR142-3p levels in PBMCs of H patients, which should consequently be linked to the suppression of IL-6 and IL-10 levels. This argues against a causal role of the miRNA for the differences in inflammatory activation in this situation. However, it cannot be excluded, at the moment, that the translation of “high IL-10 mRNA expression rates” in H patients is attenuated by degradative mechanisms exerted by miR-142-3p.

In summary, receptors for IL-10 (IL-10R) and IL-6 (IL-6R) transduce their signals after ligation via the STAT3 pathway. Our data show that a “diminished” STAT3 phosphorylation state at position S727 may change STAT3 signaling in hypervolemic HD patients. The IL-10R appears to be unique in generating potent anti-inflammatory responses, while inflammatory responses via IL-6R are largely triggered. The expression of SOCS3 is induced via both cytokine receptors. Although SOCS3, the feed-back inhibitor of IL-6, is elevated in H patients, the corresponding cytokine level remains high in immune cells of H patients. Additionally, the anti-inflammatory route seems inefficiently active. A disturbed post-translational mechanism of IL-10 appears as part of this observation. Expression of miR-143, a micro-RNA whose expression is inversely correlated with both IL-6 and IL-10 levels, is up-regulated in H patients. How far miR-142-3p expression affects IL-6 and IL-10 regulation in hypervolemia is the subject of future studies.

### Limitations of the Study

We conducted a cross-sectional pilot study to detect abnormalities in signaling pathways controlling both inflammatory and anti-inflammatory activations in hypervolemic HD patients. The study is limited by the small sample size; however, our data provide relevant information for the design of future studies.

The main limitation of our study is that the volume status of our patients determined by bioimpedance analysis was not substantiated with other methods. A multiparametric approach may give—depending on the method used—useful information about the exact nature of the disease. A key limitation of BIA is its inability to detect the location of extracellular volume expansion. Thus, different methods, such as point of care ultrasonography (POCUS), focused cardiac ultrasound (FoCUS), venous excess ultrasound (VExUS), lung ultrasound (LUS), or even chest radiography, will support bioimpedance analysis. Also, the analysis of specific biomarkers, such as carbohydrate antigen 125 (CA125) and pro-brain natriuretic peptide (pro-BNP), may be helpful in the exact diagnosis of overhydrated patients [43].

A further essential drawback is the observational nature of our study. We can only state that STAT3 signaling and miR-142-3p expression are different between normo- and hypervolemic HD patients, while proof of causality can only be derived from interventional studies.

## 4. Materials and Methods

### 4.1. Study Population

The study population was described recently [21]. In short, there were 40 patients on maintenance hemodialysis at the Nephrology Outpatient Dialysis Center of the Department of Internal Medicine II of the University Halle-Wittenberg were enrolled in the cross-sectional study. Inclusion criteria included age > 18 years, a history of hemodialysis treatment > 12 weeks, and informed consent. Patients with active malignancy, active infections, and neurological disorders were excluded from the study. Fluid status and bioelectrical impedance vector analysis [44] were performed using the Haemo-Master (Nikkiso-DBB-EXA) (Nikkiso Medical Europe, Hamburg, Germany), a multi-frequency bioimpedance device. Only well-nourished patients without pacemakers, metal-on-metal prostheses, decompensated cirrhosis, and without amputations were included in the study. Regarding the dialysis-specific parameters, most of the patients (N = 39) were dialyzed using high-flux dialyzers (N = 39), and 1 patient was treated with a low-flux dialyzer. The dialysis vintage was comparable in both groups (N: 7.6 ± 5.7 vs. 7.1 ± 10.9 years). The conventional dialysis sessions were performed using poly-sulfone (N = 39) and cellulose membranes (N = 1). The residual rest diuresis was also comparable in both groups (N: 491 ± 771 vs. H: 465 ± 468 mL). Eleven (26%) patients were dialyzed through permanent catheters, twenty-six (62%) patients were dialyzed through arteriovenous fistulas, and five (12%) patients had an arteriovenous graft. The reasons for kidney failure were glomerulonephritis (17.5%), diabetic nephropathy (25.0%), interstitial nephritis (2.5%), ischemic nephropathy (17.5%), and others (37.5%).

All immunobiological samples were taken after the long intradialytic interval and before the start of the first dialysis session of the week. Normohydration was defined as the range of impedance vectors falling within the 75% confidence interval of the reference data provided by the manufacturer, and overhydration (hypervolemia) was defined by vectors > 75th percentile. The study was conducted according to the Declaration of Helsinki. Written informed consent was obtained from all study subjects, and the study protocol was approved by the local ethics committee.

### 4.2. PBMC Isolation

PBMCs were isolated by Ficoll density centrifugation (GE Healthcare, Solingen, Germany) from EDTA blood samples drawn from the dialysis fistula before the dialysis session. The quality of isolated cells was tested by 7-AAD staining Thermo Fisher Scientific, Darmstadt, Germany). The vitality of PBMC was 99.2% ± 1.6 for N and 99.1% ± 0.5 for H patients.

### 4.3. RNA/cDNA/qPCR

RNA was isolated from PBMC lysates using the Quick RNA MiniPrep Isolation Plus Kit (ZymoResearch, Freiburg, Germany). The RNA concentration and quality (260/280 ratio: 1.8 ± 0.1) were tested using the Nanodrop technique (PEQLAB Biotechnologie GmbH, Erlangen, Germany). Equal amounts of RNA (50 ng) were reverse transcribed using the High-Capacity cDNA Reverse Transcription Kit (Thermo Fisher Scientific).

IL-10 (Hs00961622_m1), Cyp1A (Hs01052496_g1), Cyp1B (Hs00164383_m1), IL-6 (Hs00985639_m1), SOCS1 (HS00705164_s1), SOCS3 (HS02330326_s1), and RPL37A (Hs01102345_m1) mRNA expressions were analyzed using TaqMan probes (Thermo Fisher Scientific) using qPCRBIO Probe Mix High-ROX (Nippon Genetics, Düren, Germany). The samples were processed in duplicates on a StepOnePlus Cycler (Thermo Fisher Scientific). Data were normalized by RPL37A and related to healthy control donor RNA. Thus, results were expressed as the x-fold difference compared to the healthy control.

### 4.4. Analysis of IL-10 Secretion by Flow Cytometry

The 0.5 × 10^6^ PBMCs were incubated under unstimulated and stimulated conditions (LPS (0127:B8, 25 ng/mL (Sigma-Aldrich, Steinheim, Germany) or LPS (25 ng/mL)/AhR-A (2 µg/mL, Merck Millipore, Darmstadt, Germany)). Samples were incubated in the presence of monensin (end concentration: 0.5-fold, BioLegend, Amsterdam, The Netherlands). After 14 h, cells were stained with the viability dye (405/450, Miltenyi Biotec, Bergisch-Gladbach, Germany), followed by the surface staining of cells with anti-CD14 and anti-CD3 (both Miltenyi Biotec). The cells were fixed using 1% paraformaldehyde solution. After saponin treatment, PBMCs were stained with anti-IL-10 (Miltenyi Biotec) or a corresponding isotype control (BioLegend). Samples were analyzed on the MACS Quant analyzer (Miltenyi Biotec) using MACS Quantify software Version 2.8.

### 4.5. Analysis of STAT3 (pY705) and STAT3 (pS727) by Phospho-Flow Cytometry

The 0.5 × 10^6^ PBMCs were incubated under unstimulated and stimulated conditions (LPS (0127:B8, 25 ng/mL (Sigma-Aldrich) or LPS (25 ng/mL)/AhR-A (2 µg/mL, Merck Millipore)). After 30 min, the cells were fixed using 4% paraformaldehyde solution, followed by the permeabilization of cells using ice-cold PhosFlow Perm Buffer III (BD Biosciences, Heidelberg, Germany). Afterwards, cells were stained with anti-CD86, anti-CD4 (both Thermo Fisher Scientific), anti-phospho-STAT3 ((Tyr705), (STAT3 pY705), (BioLegend)), and anti-phospho-STAT3 ((Ser727), STAT3 pS727, BD Biosciences). Samples were analyzed on the MACS Quant analyzer (Miltenyi Biotec) using MACS Quantify software. Gates were set according to FMO controls.

### 4.6. Analysis of Phospho-4E-BP1 by Phospho-Flow Cytometry

The 0.5 × 10^6^ PBMCs were incubated under unstimulated and stimulated conditions (LPS (0127:B8, 25 ng/mL (Sigma-Aldrich) or LPS (25 ng/mL)/AhR-A (2 µg/mL, Merck Millipore)). After 30 min, the cells were fixed using 4% paraformaldehyde solution, followed by the permeabilization of cells using ice-cold PhosFlow Perm Buffer III (BD Biosciences, Heidelberg, Germany). Afterwards, cells were stained with anti-CD86, anti-CD4 (both Thermo Fisher Scientific), and anti-phospho-4E-BP1 (Thr37, 46; p4E-BP1) (ThermoFisher Scientific). Samples were analyzed on the MACS Quant analyzer (Miltenyi Biotec) using MACS Quantify software. Gates were set according to FMO controls.

### 4.7. miR-142-3p/5p Analysis

RNA was isolated from PBMC lysates using the Quick RNA MiniPrep Isolation Plus Kit (ZymoResearch, Freiburg, Germany). A miRNeasy serum spike-in miR-39 miRNA (Qiagen, Hilden, Germany) was integrated in the micro-RNA isolation procedure. Equal amounts of RNA (10ng) were reverse transcribed using the Advanced miRNA cDNA Synthesis Kit (Thermo Fisher Scientific). cDNA samples were diluted 1:60, followed by qPCR using the miRCURY LNA SYBR Green PCR Kit (Qiagen). Primers used for the detection of Hsa-142-3p/5p (YP00204291/YP00204722) were purchased from Qiagen. The samples were processed in duplicates on a StepOnePlus Cycler (Thermo Fisher Scientific). Data were normalized by miR-39 and related to healthy control donor miR. Thus, results were expressed as the x-fold difference compared to the healthy control.

### 4.8. Cytokine Analysis

IL-10 serum levels were analyzed using high-sensitivity ELISAs (Tecan, Crailsheim, Germany). IL-6 was measured in supernatants of PBMCs using the Legend MAX^TM^ ELISA kit for IL-6 (BioLegend). Data were analyzed on the ELX808 microplate reader (Bio-Tek Inc., Berlin, Germany).

### 4.9. Statistics

Results were expressed as mean ± SD. All continuous variables were controlled for normal distribution using the D’Agostino–Pearson omnibus test. Continuous data were compared using the Mann–Whitney test or using one-way ANOVA, followed by the Friedman post-test as appropriate. All calculations were carried out using the SPSS 21.0 (SPSS Inc., Chicago, IL, USA) or GraphPad Prism 9.2.0 statistics software (GraphPad Software Inc., La Jolla, CA, USA). The level of significance was set at *p* < 0.05.

## Figures and Tables

**Figure 1 ijms-25-03719-f001:**
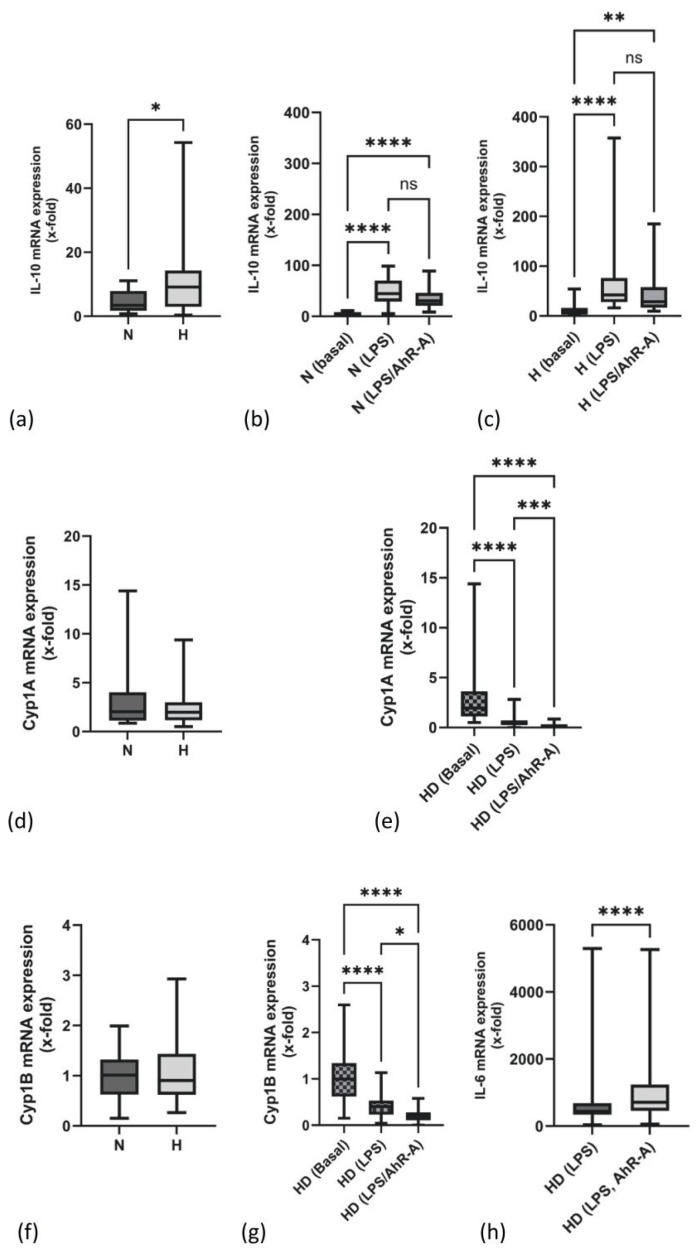
IL-10 mRNA expression in unstimulated (**a**) and LPS- or LPS/AhR-A-stimulated PBMCs of normo- (N, (**b**)) and hypervolemic (H, (**c**)) patients. Comparison of Cyp1A mRNA expression in unstimulated PBMCs of N and H patients (**d**). Influence of LPS or LPS/AhR stimulation on Cyp1A expression in the total hemodialysis (HD) cohort (**e**). Analysis of Cyp1B mRNA expression in unstimulated PBMCs of N and H patients (**f**). Effect of LPS or LPS/AhR stimulation on Cyp1B expression in HD patients (**g**). Analysis of IL-6 mRNA expression upon LPS versus LPS/AhR-A challenge in HD patients (**h**). Data are presented as box plots with the median and the 25/75 percentile. Data were analyzed using the Mann–Whitney test or the one way ANOVA as appropriate. * *p* < 0.05, ** *p* < 0.01, *** *p* < 0.001, and **** *p* < 0.0001. ns: not significant.

**Figure 2 ijms-25-03719-f002:**
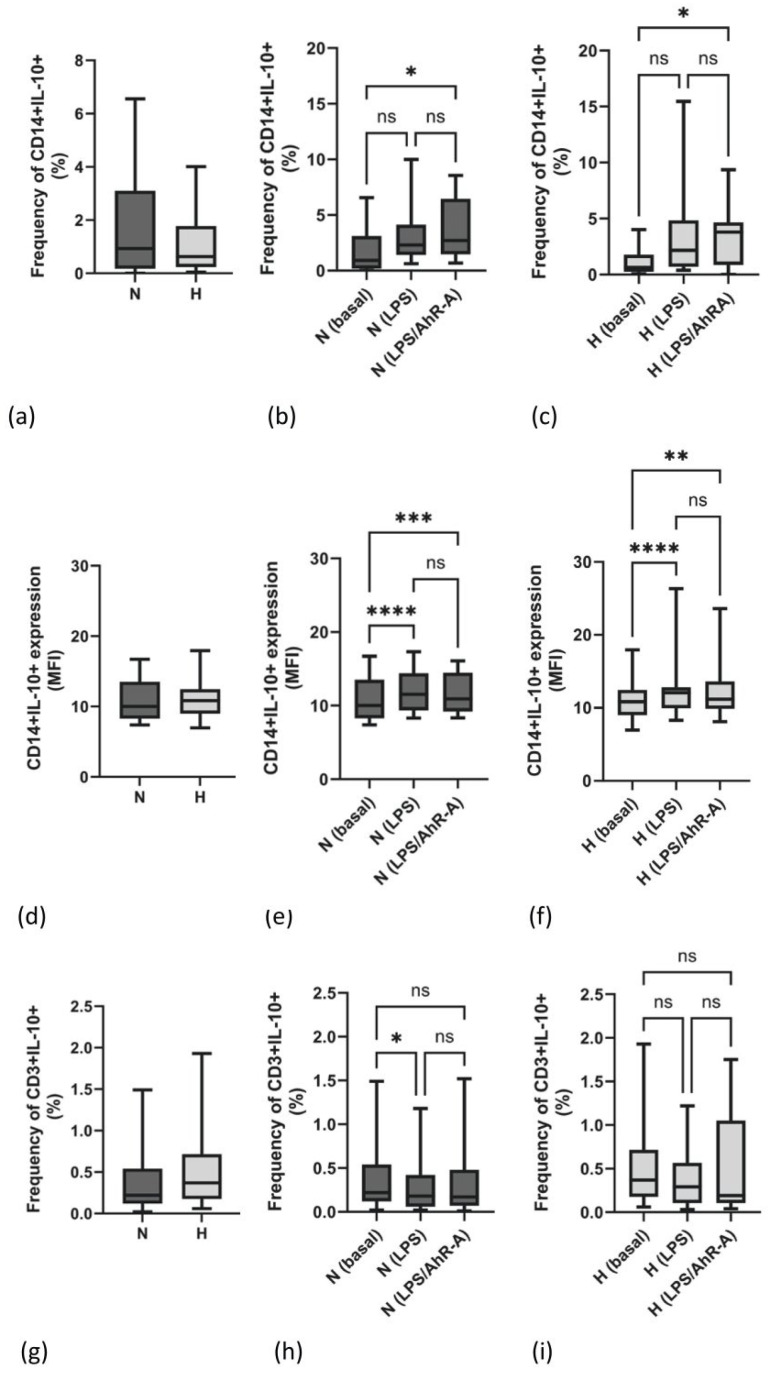
The percentage of monocytes (CD14+) staining positive for IL-10 in normo- (N) and hypervolemic (H) patients (**a**). (**b**,**c**) present the frequency of monocytes upon stimulation with LPS or LPS/AhR-A in N and H patients. The expression density per cell (MFI) in monocytes of N and H patients is given in (**d**). Monocytic IL-10 expression under basal and stimulated conditions (LPS, LPS/AhR-A) in N (**e**) and H (**f**) patients. (**g**) illustrates the percentage of lymphocytes (CD3+) staining positive for IL-10 in N and H patients. The percentage of IL-10-expressing cells under basal and stimulated conditions (LPS, LPS/AhR-A) is given in (**h**) (N patients) and (**i**) (H patients). Data are presented as box plots with the median and the 25/75 percentile. Data were analyzed using the Mann–Whitney test or one way ANOVA as appropriate. * *p* < 0.05, ** *p* < 0.01, *** *p* < 0.001, and **** *p* < 0.0001. ns: not significant.

**Figure 3 ijms-25-03719-f003:**
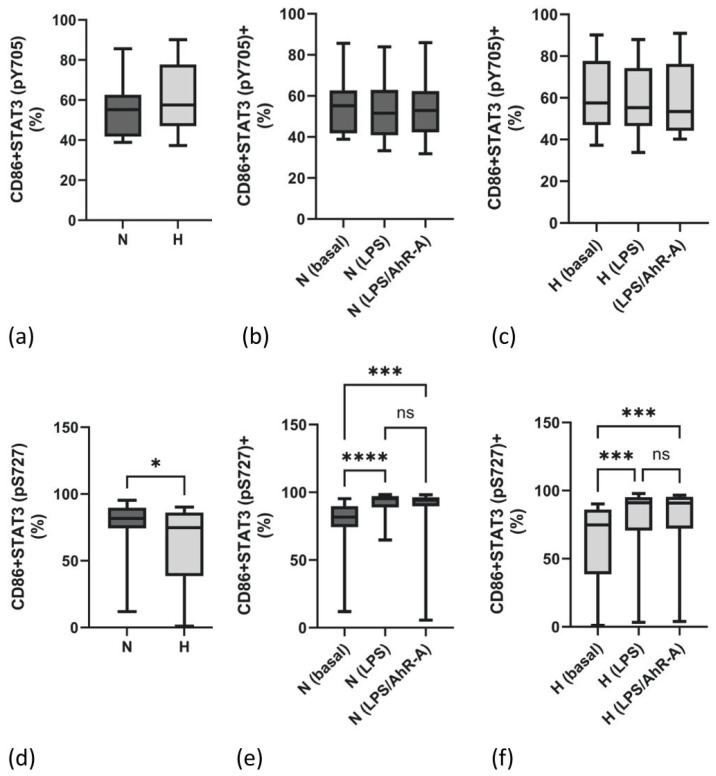
Frequency of monocytes (CD86+) staining positive for the phosphorylated form of STAT3 at position 705 (pY705) in normo- (N) and hypervolemic (H) patients (**a**). (**b**,**c**) present the frequency of STAT3 (pY705)-positive monocytes upon stimulation with LPS or LPS/AhR-A in N and H patients. (**d**) illustrates the percentage of monocytes staining positive for the phosphorylated form of STAT3 at position 727 (pS727) in N and H patients. The percentage of STAT3 (pS727)-expressing cells under basal and stimulated conditions (LPS, LPS/AhR-A) is given in (**e**) (N patients) and (**f**) (H patients). Data are presented as box plots with the median and the 25/75 percentile. Data were analyzed using the Mann–Whitney test or one way ANOVA as appropriate. * *p* < 0.05, *** *p* < 0.001, and **** *p* < 0.0001. ns: not significant.

**Figure 4 ijms-25-03719-f004:**
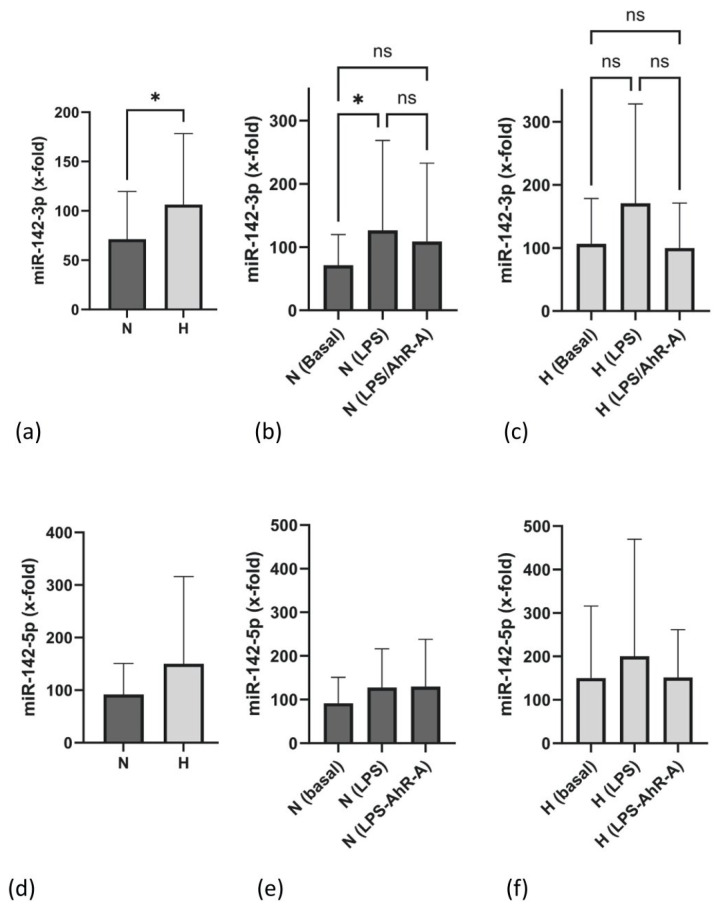
Expression of miR-142-3p in PBMCs of normo- (N) and hypervolemic (H) patients (**a**). (**b**,**c**) present the expression of miR-142-3p upon stimulation with LPS or LPS/AhR-A in N and H patients. (**d**) illustrates the expression of miR-142-5p in N and H patients. The expression of miR-142-5p under basal and stimulated conditions (LPS, LPS/AhR-A) is given in (**e**) (N patients) and (**f**) (H patients). Data are presented as box plots with the median and the 25/75 percentile. Data were analyzed using the Mann–Whitney test or one way ANOVA as appropriate. * *p* < 0.05. ns: not significant.

**Table 1 ijms-25-03719-t001:** Frequency of monocytes staining positive for the phosphorylated form of the translational repressor-binding protein 4E-BP1 (p4E-BP1).

	%CD86+ p4E-PB1 (Basal)	%CD86+ p4E-PB1 (LPS)	%CD86+ p4E-PB1 (LPS/AhR)
N	2.0 [0.7–13.2]	13.4 [3.3–65.6]	13.2 [4.3–65.6]
H	3.1 [0.4–24.2]	9.6 [2.4–88.9]	13.8 [3.1–80.7]
*p* value	*p* = 0.125	*p* = 0.823	*p* = 0.730

Data are expressed as the median [range: min.–max.].

**Table 2 ijms-25-03719-t002:** Frequency of CD4+ lymphocytes staining positive for the phosphorylated form of the translational repressor binding protein 4E-BP1 (p4E-BP1).

	%CD4+ p4E-PB1 (Basal)	%CD4+ p4E-PB1 (LPS)	%CD4+ p4E-PB1 (LPS/AhR)
N	1.7 [0.4–83.5]	2.4 [0.5–92.2]	2.3 [0.7–90.4]
H	3.2 [0.04–94.0]	3.9 [0.3–97.5]	3.7 [0.2–97.1]
*p* value	*p* = 0.261	*p* = 0.569	*p* = 0.628

Data are expressed as the median [range: min.–max.].

**Table 3 ijms-25-03719-t003:** SOCS 1 and SOC3 transcripts in normo- (N) and hypervolemic patients (H).

	SOCS1 mRNA Expression	SOCS3 mRNA Expression
N	1.7 [0.4–5.4]	2.2 [1.0–4.8]
H	2.0 [0.4–3.5]	3.3 [0.6–9.4]
*p*-value	0.575	0.040

Data are expressed as the median [range: min.–max.].

## Data Availability

Data is contained within the article.

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
