# Peer review of "Hypervolemia in Dialysis Patients Impairs STAT3 Signaling and Upregulates miR-142-3p: Effects on IL-10 and IL-6"

_ijms, 2024, doi:10.3390/ijms25073719_

Round 1
Reviewer 1 Report
Comments and Suggestions for Authors
First of all, we would like to thank the authors for this original work that aims to describe translational and post-translational mechanisms of cytokine induction in overhydrated patients on hemodialysis. Interesting work at present.
First: Although BIA is assumed to be the Gold Standard for measuring volume overload in hemodialysis patients, it has many limitations that need to be discussed in the methodology. One of them is that it excludes patients with pacemakers, mmii metal prostheses, decompensated cirrhosis and amputees. It will be necessary to describe that there are no patients with these characteristics in the study. Furthermore, this procedure is not validated for malnourished patients as well as those with obesity (BMI below 16 and BMI above 34 kg/m2). I am adding these bibliographic citations to be included in the commentary on the discussion and methodology.
-Mayne et al. Bioimpedance in CKD: an untapped resource?. NDT 2023, odir:10.1093/ndt/gfac275
--Wang, Y.; Gu, Z. Effect of bioimpedance-defined overhydration parameters on mortality and cardiovascular events in patients undergoing dialysis: A systematic review and meta-analysis. J. Int. Med. Res. 2021
Second, and I think the most important, currently only basing the diagnosis of fluid overload in patients on hemodialysis with BIA is very limiting, knowing that this is a multiparametric view (serum biomarkers with Ca125, proBNP, POCUS, LUS, VExUS... FOCUS). To comment on this in the discussion and as an important limitation of the study.
-Wang, Y.; Gu, Z. Effect of bioimpedance-defined overhydration parameters on mortality and cardiovascular events in patients undergoing dialysis: A systematic review and meta-analysis. J. Int. Med. Res. 2021
Third, why didn't they use another device such as the Freseinus BCM which is validated in hemodialysis?
Mayne et al. BIA in CKD: an untapped resource? NDT 2023.
In the materials and methods section, in the study population, add hemodialysis data: Type of HD (online, conventional, expand, high or low flow), dialysis vintage, dialysis frequency (incremental vs conventional), type of dialzer, if patients had residual renal function, type of vascular access (native fistule vs tunneled catheter).
Also describe the type of device with which the BIA measurements are made, whether it is single or multifrequency.
Comments on the Quality of English LanguageMinor editing of English language required
Author Response
We would like to thank the reviewer for his helpful comments!
Response to Reviewer 1 Comments
Point 1: The Reviewer states that although BIA is assumed to be the Gold Standard for measuring volume overload in hemodialysis patients, it has many limitations that need to be discussed. Patients with pacemakers, mmii metal prostheses, decompensated cirrhosis and amputees as well as malnourished patients should be excluded from BIA analysis.
Response 1: We thank the Reviewer for these useful suggestions. We added the following sentences to the Materials and Method section (4.1 Study Population): “Only well-nourished patients without pacemakers, metal on meta prosthesis, decompensated cirrhosis and without amputations were included in the study.” Lines 345-347.
“In spite of different limitations (patients with pacemakers, amputees, patients with metal on metal prosthesis as well as decompensated cirrhosis patients have to be excluded from analysis), bioimpedance analysis is assumed the Gold standard for measuring volume overload in hemodialysis patients (Mayne et al. 2023). Several devices for monitoring fluid overload have been investigated in HD patients. These includes the Fresenius Body Composition Monitor (Mayne et al. 2023), Nikkiso DBB device (Antlanger et al. 2017), and the Nutriguard-M device (Beberashvili et al. 2018). The importance of bioimpedance analysis in dialysis patients is highlighted by the fact that bioimpedance parameters most probably represent independent predictors for mortality and cardiovascular events (Wang and Gu 2021).”
This information was added to the Discussion section. Lines 219-227.
The corresponding references were added to the Reference section:
Ref. 22: Mayne, Kaitlin J.; Lees, Jennifer S.; Herrington, William G. (2023): Bioimpedance in CKD: an untapped resource? In Nephrology, dialysis, transplantation: official publication of the European Dialysis and Transplant Association - European Renal Association 38 (3), pp. 583–585. DOI: 10.1093/ndt/gfac275.
Ref. 23: Antlanger, Marlies; Josten, Peter; Kammer, Michael; Exner, Isabella; Lorenz-Turnheim, Katharina; Eigner, Manfred et al. (2017): Blood volume-monitored regulation of ultrafiltration to decrease the dry weight in fluid-overloaded hemodialysis patients: a randomized controlled trial. In BMC nephrology 18 (1), p. 238. DOI: 10.1186/s12882-017-0639-x.
Ref. 24: Beberashvili, Ilia; Yermolayeva, Tatyana; Katkov, Anna; Garra, Nedal; Feldman, Leonid; Gorelik, Oleg et al. (2018): Estimating of Residual Kidney Function by Multi-Frequency Bioelectrical Impedance Analysis in Hemodialysis Patients Without Urine Collection. In Kidney & blood pressure research 43 (1), pp. 98–109. DOI: 10.1159/000487106.
Ref. 25: Wang, Yinghui; Gao, Lu (2022): Inflammation and Cardiovascular Disease Associated with Hemodialysis for End-Stage Renal Disease. In Frontiers in pharmacology 13, p. 800950. DOI: 10.3389/fphar.2022.800950.
Point 2: The Reviewer points out that the analysis of fluid overload in patients is a very limiting method which does not allow insight in the pathophysiology of the underlying disease.
Response 2: The Reviewer is right. “It is a main limitation of our study that the volume status of our patients determined by bioimpedance analysis was not substantiated with other methods. A multiparametric approach may give – in dependence of the method used – useful information about the exact nature of the disease. For a key limitation of BIA is its inability to detect the location of extracellular volume expansion. Thus, different methods such as point of care ultrasonography (POCUS), focused cardiac ultrasound (FoCUS), venous excess ultrasound (VExUS), lung ultrasound (LUS) or even chest radiography will support bioimpedance analysis. Also, the analysis of specific biomarkers such as carbohydrate antigen 125 (CA125) and pro brain natriuretic peptide (pro-BNP) may be helpful in the exact diagnosis of overhydrated patients (Koratala et al. 2022).”
These sentences were added to the Limitation of the Study Section. Lines 322-331.
The corresponding reference was added to the Reference section:
Ref. 43: Koratala, Abhilash; Ronco, Claudio; Kazory, Amir (2022): Diagnosis of Fluid Overload: From Conventional to Contemporary Concepts. In Cardiorenal medicine 12 (4), pp. 141–154. DOI: 10.1159/000526902.
Point 3: The Reviewer asks why we did not use the Fresenius BCM device for analysis of overhydration in HD.
Response 3: The patients were recruited from the Nephrology outpatient dialysis center of the Department of Internal Medicine II at the University of Halle-Wittenberg. Unfortunately, the outpatient center has only access to Nikkiso devices but not to the Fresenius Body Composition Monitor.
Point 4: The Reviewer is of the opinion that some patient characteristics must be added to the method section: Type of HD (online, conventional, expand, high or low flow), dialysis vintage, dialysis frequency (incremental vs conventional), type of dialyzer, if patients had residual renal function, type of vascular access (native fistula vs tunneled catheter).
Response 4: We agree with the Reviewer and added the following sentences to the Materials and Method section (4.1 Study population): “Regarding the dialysis specific parameters, most of the patients (N=39) were dialysed using high-flux dialyzers (N=39), 1 patient was treated with low-flux dialyzer. The dialysis vintage was comparable in both groups (N: 7.6±5.7 vs. 7.1±10.9 years). The conventional dialysis sessions were performed using poly sulfone (N=39) and cellulose membranes (N=1). The residual rest diuresis was also comparable in both groups (N. 491±771 vs H: 465±468 ml). Eleven (26%) patients were dialyzed through permanent catheters, 26 (62%) patients through arteriovenous fistulas and 5 (12%) patients had an arteriovenous graft. Lines 347-354.
Point 5: The Reviewer wants to know which device was used for BIA measurement and whether it is single or multifrequency.
Response 5: Using the Haemo-Master (Nikkiso-DBB-EXA) device we investigated the hydration status of HD patients via multi-frequency bioimpedance measurements.
This information was added to the Materials and Method section (4.1 Study Population). Lines 344-345.

Reviewer 2 Report
Comments and Suggestions for Authors
This is a cross-sectional pilot study. The authors enrolled 40 hemodyalisis patients and divided them into hypervolemic and normovolemic group to evaluate possible abnormalities in signaling pathawys involved in inflammatory and antiinflammatory activation in hypervolemic patients.
This research is interesting and obtained results contain useful information for performing further more extensive studies.
I have only some minor comments.
The authors have already published some results on the same study population, and cited ref. 21. Maybe, it is appropriate to include a little more information about the patients, in present manuscript, at least the kidney diseases that were the causes for hemodyalisis...
The authors should comment their previous results performed on the same study population. Are obtained data in accordance with results from previuos research (ref 21)?
The abbrevation PBMCs should be explained, the first time it is mentioned in the text.
Comments on the Quality of English LanguageMinor editing of English language required.
Author Response
We would like to thank the reviewer for his helpful comments!
Response to Reviewer 2 Comments
Point 1: The Reviewer is of the opinion that some patient-relevant characteristics (including the cause of kidney disease) should be added to the method section.
Response 1: The Reviewer is right we added the following sentences to the Materials and Method section (4.1 Study population): “Regarding the dialysis specific parameters, most of the patients (N=39) were dialysed using high-flux dialyzers (N=39), 1 patient was treated with low-flux dialyzer. The dialysis vintage was comparable in both groups (N: 7.6±5.7 vs. 7.1±10.9 years). The conventional dialysis sessions were performed using poly sulfone (N=39) and cellulose membranes (N=1). The residual rest diuresis was also comparable in both groups (N. 491±771 vs H: 465±468 ml). Eleven (26%) patients were dialyzed through permanent catheters, 26 (62%) patients through arteriovenous fistulas and 5 (12%) patients had an arteriovenous graft. The reasons for kidney failure were glomerulonephritis (17.5%), diabetic nephropathy (25.0%), interstitial nephritis (2.5%), ischaemic nephropathy (17.5%) and others (37.5%).” Lines 347-356.
Point 2: The Reviewer some data of the study cohort have been published. He wants to know if the data obtained are in accordance of the results with the previous research (Ulrich et al. IJMS 2023).
Response 2: The Reviewer is right the data obtained were collected from the same patients as described in Ulrich et al, 2023. So, of course, the patient characteristics are the same in both studies. However, the issue of this study involves the relevance of STAT3 signaling in hypervolemic HD patients and its sequelae on IL-10 and IL-6 production, whereas the former study reported the influence of hypervolemia on NF-kB signaling and caspase-4 induction. Thus, there are two quite different results in both papers which cannot be compared.
Point 3: The Reviewer remarks that the abbreviation of the term PBMCs should be explained the first time it is mentioned in the text.
Response 3: We apologize for this mistake and added the explanation (“peripheral blood mononuclear cells”). Lines 15-16.

Round 2
Reviewer 1 Report
Comments and Suggestions for Authors
Thank the authors for having accepted our recommendations for improving the manuscript. On my part, after including the necessary data according to my recommendations, the article can be published.